# Auxiliary Task Reweighting for Minimum-data Learning

**Baifeng Shi**
Peking University
bfshi@pku.edu.cn

**Judy Hoffman**
Georgia Institute of Technology
judy@gatech.edu

**Kate Saenko**
Boston University & MIT-IBM Watson AI Lab
saenko@bu.edu

**Trevor Darrell, Huijuan Xu**
University of California, Berkeley
{trevor, huijuan}@eecs.berkeley.edu

## Abstract

Supervised learning requires a large amount of training data, limiting its application where labeled data is scarce. To compensate for data scarcity, one possible method is to utilize auxiliary tasks to provide additional supervision for the main task. Assigning and optimizing the importance weights for different auxiliary tasks remains an crucial and largely understudied research question. In this work, we propose a method to automatically reweight auxiliary tasks in order to reduce the data requirement on the main task. Specifically, we formulate the weighted likelihood function of auxiliary tasks as a surrogate prior for the main task. By adjusting the auxiliary task weights to minimize the divergence between the surrogate prior and the true prior of the main task, we obtain a more accurate prior estimation, achieving the goal of minimizing the required amount of training data for the main task and avoiding a costly grid search. In multiple experimental settings (*e.g.* semi-supervised learning, multi-label classification), we demonstrate that our algorithm can effectively utilize limited labeled data of the main task with the benefit of auxiliary tasks compared with previous task reweighting methods. We also show that under extreme cases with only a few extra examples (*e.g.* few-shot domain adaptation), our algorithm results in significant improvement over the baseline. Our code and video is available at https://sites.google.com/view/auxiliary-task-reweighting.

## 1 Introduction

Supervised deep learning methods typically require an enormous amount of labeled data, which for many applications, is difficult, time-consuming, expensive, or even impossible to collect. As a result, there is a significant amount of research effort devoted to efficient learning with limited labeled data, including semi-supervised learning [41, 47], transfer learning [48], few-shot learning [9], domain adaptation [49], and representation learning [42].

Among these different approaches, auxiliary tasks are widely used to alleviate the lack of data by providing additional supervision, *i.e.* using the same data or auxiliary data for a different learning task during the training procedure. Auxiliary tasks are usually collected from related tasks or domains

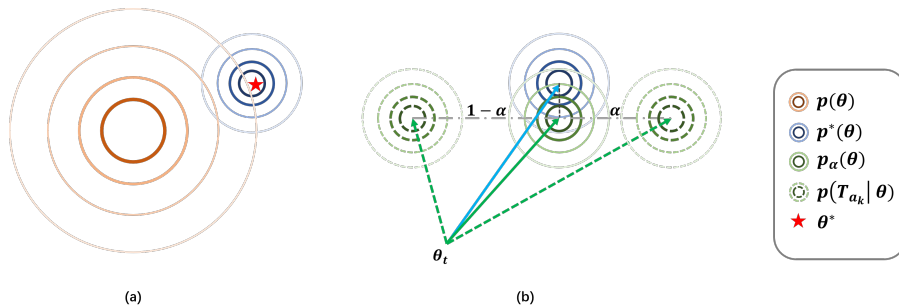

Figure 1: Learning with minimal data through auxiliary task reweighting. (a) An ordinary prior $p(\theta)$ over model parameters contains little information about the true prior $p^*(\theta)$ and the optimal parameter $\theta^*$. (b) Through a weighted combination of distributions induced by data likelihood $p(\mathcal{T}_{a_k}|\theta)$ of different auxiliary tasks, we find the optimal surrogate prior $p_\alpha(\theta)$ which is closest to the true prior.

where there is abundant data [49], or manually designed to fit the latent data structure [46, 55]. Training with auxiliary tasks has been shown to achieve better generalization [2], and is therefore widely used in many applications, *e.g.* semi-supervised learning [55], self-supervised learning [42], transfer learning [48], and reinforcement learning [25].

Usually both the main task and auxiliary task are jointly trained, but only the main task's performance is important for the downstream goals. The auxiliary tasks should be able to reduce the amount of labeled data required to achieve a given performance for the main task. However, this has proven to be a difficult selection problem as certain seemingly related auxiliary tasks yield little or no improvement for the main task. One simple task selection strategy is to compare the main task performance when training with each auxiliary task separately [54]. However, this requires an exhaustive enumeration of all candidate tasks, which is prohibitively expensive when the candidate pool is large. Furthermore, individual tasks may behave unexpectedly when combined together for final training. Another strategy is training all auxiliary tasks together in a single pass and using an evaluation technique or algorithm to automatically determine the importance weight for each task. There are several works along this direction [4, 10, 13, 31], but they either only filter out unrelated tasks without further differentiating among related ones, or have a focused motivation (*e.g.* faster training) limiting their general use.

In this work, we propose a method to adaptively reweight auxiliary tasks on the fly during joint training so that the data requirement on the main task is minimized. We start from a key insight: *we can reduce the data requirement by choosing a high-quality prior*. Then we formulate the parameter distribution induced by the auxiliary tasks' likelihood as a surrogate prior for the main task. By adjusting the auxiliary task weights, the divergence between the surrogate prior and the true prior of the main task is minimized. In this way, the data requirement on the main task is reduced under high quality surrogate prior. Specifically, due to the fact that minimizing the divergence is intractable, we turn the optimization problem into minimizing the distance between gradients of the main loss and the auxiliary losses, which allows us to design a practical, light-weight algorithm. We show in various experimental settings that our method can make better use of labeled data and effectively reduce the data requirement for the main task. Surprisingly, we find that very little labeled data (*e.g.* 1 image per class) is enough for our algorithm to bring a substantial improvement over unsupervised and few-shot baselines.

## 2 Learning with Minimal Data through Auxiliary Task Reweighting

Suppose we have a main task with training data $\mathcal{T}_m$ (including labels), and $K$ different auxiliary tasks with training data $\mathcal{T}_{a_k}$ for $k$-th task, where $k = 1, \cdots, K$. Our model contains a shared backbone with parameter $\theta$, and different heads for each task. Our goal is to find the optimal parameter $\theta^*$ for the main task, using data from main task as well as auxiliary tasks. Note that we care about performance on the main task and auxiliary tasks are only used to help train a better model on main task (*e.g.* when we do not have enough data on the main task). In this section, we discuss how to learn with minimal data on main task by learning and reweighting auxiliary tasks.

## 2.1 How Much Data Do We Need: Single-Task Scenario

Before discussing learning from multiple auxiliary tasks, we first start with the single-task scenario. When there is only one single task, we normally train a model by minimizing the following loss:

$$\mathcal{L}(\theta) = -\log p(\mathcal{T}_m|\theta) - \log p(\theta) = -\log(p(\mathcal{T}_m|\theta) \cdot p(\theta)), \tag{A1}$$

where $p(\mathcal{T}_m|\theta)$ is the likelihood of training data and $p(\theta)$ is the prior. Usually, a relatively weak prior (*e.g.* Gaussian prior when using weight decay) is chosen, reflecting our weak knowledge about the true parameter distribution, which we call 'ordinary prior'. Meanwhile, we also assume there exists an unknown 'true prior' $p^*(\theta)$ where the optimal parameter $\theta^*$ is actually sampled from. This true prior is normally more selective and informative (*e.g.* having a small support set) (See Fig. 1(a)) [5].

Now our question is, how much data do we need to learn the task. Actually the answer depends on the choice of the prior $p(\theta)$. If we know the informative 'true prior' $p^*(\theta)$, only a few data items are needed to localize the best parameters $\theta^*$ within the prior. However, if the prior is rather weak, we have to search $\theta$ in a larger space, which needs more data. Intuitively, the required amount of data is related to the divergence between $p(\theta)$ and $p^*(\theta)$: the closer they are, the less data we need.

In fact, it has been proven [5] that the expected amount of information needed to solve a single task is

$$\mathcal{I} = D_{\mathrm{KL}}(p^* \parallel p) + H(p^*), \tag{A2}$$

where $D_{\mathrm{KL}}(\cdot \parallel \cdot)$ is Kullback–Liebler divergence, and $H(\cdot)$ is the entropy. This means we can reduce the data requirement by choosing a prior closer to the true prior $p^*$. Suppose $p(\theta)$ is parameterized by $\alpha$, *i.e.*, $p(\theta) = p_\alpha(\theta)$, then we can minimize data requirement by choosing $\alpha$ that satisfies:

$$\min_\alpha D_{\mathrm{KL}}(p^* \parallel p_\alpha). \tag{A3}$$

However, due to our limited knowledge about the true prior $p^*$, it is unlikely to manually design a family of $p_\alpha$ that has a small value in (A3). Instead, we will show that we can define $p_\alpha$ implicitly through auxiliary tasks, utilizing their natural connections to the main task.

## 2.2 Auxiliary-Task Reweighting

When using auxiliary tasks, we optimize the following joint-training loss:

$$\mathcal{L}(\theta) = -\log p(\mathcal{T}_m|\theta) - \sum_{k=1}^{K} \alpha_k \log p(\mathcal{T}_{a_k}|\theta) = -\log(p(\mathcal{T}_M|\theta) \cdot \prod_{k=1}^{K} p^{\alpha_k}(\mathcal{T}_{a_k}|\theta)), \tag{A4}$$

where auxiliary losses are weighted by a set of task weights $\boldsymbol{\alpha} = (\alpha_1, \cdots, \alpha_K)$, and added together with the main loss. By comparing (A4) with single-task loss (A1), we can see that we are implicitly using $p_{\boldsymbol{\alpha}}(\theta) = \frac{1}{Z(\boldsymbol{\alpha})} \prod_{k=1}^{K} p^{\alpha_k}(\mathcal{T}_{a_k}|\theta)$ as a 'surrogate' prior for the main task, where $Z(\boldsymbol{\alpha})$ is the normalization term (partition function). Therefore, as discussed in Sec. 2.1, if we adjust task weights $\boldsymbol{\alpha}$ towards

$$\min_{\boldsymbol{\alpha}} D_{\mathrm{KL}}(p^*(\theta) \parallel \frac{1}{Z(\boldsymbol{\alpha})} \prod_{k=1}^{K} p^{\alpha_k}(\mathcal{T}_{a_k}|\theta)), \tag{A5}$$

then the data requirement on the main task can be minimized. This implies an automatic strategy of task reweighting. Higher weights can be assigned to the auxiliary tasks with more relevant information to the main task, namely the parameter distribution of the tasks is closer to that of the main task. After taking the weighted combination of auxiliary tasks, the prior information is maximized, and the main task can be learned with minimal additional information (data). See Fig. 1(b) for an illustration.

## 2.3 Our Approach

In Sec. 2.2 we have discussed about how to minimize the data requirement on the main task by reweighting and learning auxiliary tasks. However, the objective in (A5) is hard to optimize directly due to a few practical problems:

- **True Prior (P1)**: We do not know the true prior $p^*$ in advance.

- **Samples (P2)**: KL divergence is in form of an expectation, which needs samples to estimate. However, sampling from a complex distribution is non-trivial.

- **Partition Function (P3)**: Partition function $Z(\boldsymbol{\alpha}) = \int \prod_{k=1}^{K} p^{\alpha_k}(\mathcal{T}_{a_k}|\theta)d\theta$ is given by an intractable integral, preventing us from getting the accurate density function $p_{\boldsymbol{\alpha}}$.

To this end, we use different tools or approximations to design a practical algorithm, and keep its validity and effectiveness from both theoretical and empirical aspects, as presented below.

**True Prior (P1)**    In the original optimization problem (A5), we are minimizing

$$D_{\mathrm{KL}}(p^*(\theta) \parallel p_{\boldsymbol{\alpha}}(\theta)) = E_{\theta \sim p^*} \log \frac{p^*(\theta)}{p_{\boldsymbol{\alpha}}(\theta)}, \tag{A6}$$

which is the expectation of $\log \frac{p^*(\theta)}{p_{\boldsymbol{\alpha}}(\theta)}$ w.r.t. $p^*(\theta)$. The problem is, $p^*(\theta)$ is not accessible. However, we can notice that for each $\theta^*$ sampled from prior $p^*$, it is likely to give a high data likelihood $p(\mathcal{T}_m|\theta^*)$, which means $p^*(\theta)$ is 'covered' by $p(\mathcal{T}_m|\theta)$, *i.e.*, $p(\mathcal{T}_m|\theta)$ has high density both in the support set of $p^*(\theta)$, and in some regions outside. Thus we propose to minimize $D_{\mathrm{KL}}(p^m(\theta) \parallel p_{\boldsymbol{\alpha}}(\theta))$ instead of $D_{\mathrm{KL}}(p^*(\theta) \parallel p_{\boldsymbol{\alpha}}(\theta))$, where $p^m(\theta)$ is the parameter distribution induced by data likelihood $p(\mathcal{T}_m|\theta)$, *i.e.*, $p^m(\theta) \propto p(\mathcal{T}_m|\theta)$. Furthermore, we propose to take the expectation w.r.t. $\frac{1}{Z'(\boldsymbol{\alpha})} p^m(\theta) p_{\boldsymbol{\alpha}}(\theta)$ instead of $p^m(\theta)$ due to the convenience of sampling while optimizing the joint loss (see **P2** for more details). Then our objective becomes

$$\min_{\boldsymbol{\alpha}} E_{\theta \sim p^J} \log \frac{p^m(\theta)}{p_{\boldsymbol{\alpha}}(\theta)}, \tag{A7}$$

where $p^J(\theta) = \frac{1}{Z'(\boldsymbol{\alpha})} p^m(\theta) p_{\boldsymbol{\alpha}}(\theta)$, and $Z'(\boldsymbol{\alpha})$ is the normalization term.

Now we can minimize (A7) as a feasible surrogate for (A5). However, minimizing (A7) may end up with a suboptimal $\boldsymbol{\alpha}$ for (A5). Due to the fact that $p^m(\theta)$ also covers some 'overfitting area' other than $p^*(\theta)$, we may push $p_{\boldsymbol{\alpha}}(\theta)$ closer to the overfitting area instead of $p^*(\theta)$ by minimizing (A7). But we prove that, under some mild conditions, if we choose $\boldsymbol{\alpha}$ that minimizes (A7), the value of (A5) is also bounded near the optimal value:

**Theorem 1.** *(Informal and simplified version) Let us denote the optimal weights for* (A5) *and* (A7) *by $\boldsymbol{\alpha}^*$ and $\hat{\boldsymbol{\alpha}}$, respectively. Assume the true prior $p^*(\theta)$ has a small support set $S$. Let $\gamma = \max_{\boldsymbol{\alpha}} \int_{\theta \notin S} p^m(\theta) p_{\boldsymbol{\alpha}}(\theta) d\theta$ be the maximum of the integral of $p^m(\theta) p_{\boldsymbol{\alpha}}(\theta)$ outside $S$, then we have*

$$D_{\mathrm{KL}}(p^* \parallel p_{\boldsymbol{\alpha}^*}) \le D_{\mathrm{KL}}(p^* \parallel p_{\hat{\boldsymbol{\alpha}}}) \le D_{\mathrm{KL}}(p^* \parallel p_{\boldsymbol{\alpha}^*}) + C\gamma^2 - C'\gamma^2 \log \gamma. \tag{A8}$$

The formal version and proof can be found in Appendix. Theorem 1 states that optimizing (A7) can also give a near-optimal solution for (A5), as long as $\gamma$ is small. This condition holds when $p^m(\theta)$ and $p_{\boldsymbol{\alpha}}(\theta)$ do not reach a high density at the same time outside $S$. This is reasonable because overfitted parameter of main task (*i.e.*, $\theta$ giving a high training data likelihood outside $S$) is highly random, depending on how we sample the training set, thus is unlikely to meet the optimal parameters of auxiliary tasks. In practice, we also find this approximation gives a robust result (Sec. 3.3).

**Samples (P2)**    To estimate the objective in (A7), we need samples from $p^J(\theta) = \frac{1}{Z'(\boldsymbol{\alpha})} p^m(\theta) p_{\boldsymbol{\alpha}}(\theta)$. Apparently we cannot sample from this complex distribution directly. However, we notice that $p^J$ is what we optimize in the joint-training loss (A4), *i.e.*, $\mathcal{L}(\theta) \propto -\log p^J(\theta)$. To this end, we use the tool of Langevin dynamics [39, 51] to sample from $p^J$ while optimizing the joint-loss (A4). Specifically, at the $t$-th step of SGD, we inject a Gaussian noise with a certain variance into the gradient step:

$$\Delta\theta_t = \epsilon_t \nabla \log p^J(\theta) + \eta_t, \tag{A9}$$

where $\epsilon_t$ is the learning rate, and $\eta_t \sim N(0, 2\epsilon_t)$ is a Guassian noise. With the injected noise, $\theta_t$ will converge to samples from $p^J$, which can then be used to estimate (A7). In practice, we inject noise in early epochs to sample from $p^J$ and optimize $\boldsymbol{\alpha}$, and then return to regular SGD once $\boldsymbol{\alpha}$ has converged. Note that we do not anneal the learning rate as in [51] because we find in practice that stochastic gradient noise is negligible compared with injected noise (see Appendix).

**Algorithm 1** ARML (**A**uxiliary Task **R**eweighting for **M**inimum-data **L**earning)

---
**Input:** main task data $\mathcal{T}_m$, auxiliary task data $\mathcal{T}_{a_k}$, initial parameter $\theta_0$, initial task weights $\boldsymbol{\alpha}$
**Parameters:** learning rate of $t$-th iteration $\epsilon_t$, learning rate for task weights $\beta$

**for** iteration $t = 1$ to $T$ **do**
   **if** $\boldsymbol{\alpha}$ has not converged **then**
      $\theta_t \leftarrow \theta_{t-1} - \epsilon_t(-\nabla \log p(\mathcal{T}_m|\theta_{t-1}) - \sum_{k=1}^{K} \alpha_k \nabla \log p(\mathcal{T}_{a_k}|\theta_{t-1})) + \eta_t$
      $\boldsymbol{\alpha} \leftarrow \boldsymbol{\alpha} - \beta\nabla_{\boldsymbol{\alpha}}\|\nabla \log p(\mathcal{T}_m|\theta_t) - \sum_{k=1}^{K} \alpha_k \nabla \log p(\mathcal{T}_{a_k}|\theta_t)\|_2^2$
      Project $\boldsymbol{\alpha}$ back into $\mathcal{A}$
   **else**
      $\theta_t \leftarrow \theta_{t-1} - \epsilon_t(-\nabla \log p(\mathcal{T}_m|\theta_{t-1}) - \sum_{k=1}^{K} \alpha_k \nabla \log p(\mathcal{T}_{a_k}|\theta_{t-1}))$
   **end if**
**end for**

---

**Partition Function (P3)**   To estimate (A7), we need the exact value of surrogate prior $p_{\boldsymbol{\alpha}}(\theta) = \frac{1}{Z(\boldsymbol{\alpha})} \prod_{k=1}^{K} p^{\alpha_k}(\mathcal{T}_{a_k}|\theta)$. Although we can easily calculate the data likelihood $p(\mathcal{T}_{a_k}|\theta)$, the partition function $Z(\boldsymbol{\alpha})$ is intractable. The same problem also occurs in model estimation [18], Bayesian inference [38], *etc*. A common solution is to use score function $\nabla \log p_{\boldsymbol{\alpha}}(\theta)$ as a substitution of $p_{\boldsymbol{\alpha}}(\theta)$ to estimate relationship with other distributions [22, 24, 33]. For one reason, score function can uniquely decide the distribution. It also has other nice properties. For example, the divergence defined on score functions (also known as *Fisher divergence*)

$$F(p \parallel q) = E_{\theta \sim p}\|\nabla \log p(\theta) - \nabla \log q(\theta)\|_2^2 \tag{A10}$$

is stronger than many other divergences including KL divergence, Hellinger distance, *etc*. [22, 32]. Most importantly, using score function can obviate estimation of partition function which is constant w.r.t. $\theta$. To this end, we propose to minimize the distance between score functions instead, and our objective finally becomes

$$\min_{\boldsymbol{\alpha}} E_{\theta \sim p^J}\|\nabla \log p(\mathcal{T}_m|\theta) - \nabla \log p_{\boldsymbol{\alpha}}(\theta)\|_2^2. \tag{A11}$$

Note that $\nabla \log p^m(\theta) = \nabla \log p(\mathcal{T}_m|\theta)$. In Appendix we show that under mild conditions the optimal solution for (A11) is also the optimal or near-optimal $\boldsymbol{\alpha}$ for (A5) and (A7) . We find in practice that optimizing (A11) generally gives optimal weights for minimum-data learning.

## 2.4   Algorithm

Now we present the final algorithm of auxiliary task reweighting for minimum-data learning (ARML). The full algorithm is shown in Alg. 1. First, our objective is (A11). Until $\boldsymbol{\alpha}$ converges, we use Langevin dynamics (A9) to collect samples at each iteration, and then use them to estimate (A11) and update $\alpha$. Additionally, we only search $\boldsymbol{\alpha}$ in an affine simplex $\mathcal{A} = \{\boldsymbol{\alpha}| \sum_k \alpha_k = K; \ \alpha_k \geq 0, \forall k\}$ to decouple task reweighting from the global weight of auxiliary tasks [10]. Please also see Appendix **??** for details on the algorithm implementation in practice.

## 3   Experiments

For experiments, we test effectiveness and robustness of ARML under various settings. This section is organized as follows. First in Sec. 3.1, we test whether ARML can reduce data requirement in different settings (semi-supervised learning, multi-label classification), and compare it with other reweighting methods. In Sec. 3.2, we study an extreme case: based on an unsupervised setting (*e.g.* domain generalization), if a little extra labeled data is provided (*e.g.* 1 or 5 labels per class), can ARML maximize its benefit and bring a non-trivial improvement over unsupervised baseline and other few-shot algorithms? Finally in Sec. 3.3, we test ARML's robustness under different levels of data scarcity and validate the rationality of approximation we made in Sec. 2.3.

## 3.1   ARML can Minimize Data Requirement

To get started, we show that ARML can minimize data requirement under two realist settings: semi-supervised learning and multi-label classification. we consider the following task reweight-

ing methods for comparison: (i) **Uniform (baseline)**: all weights are set to 1, (ii) **AdaLoss** [21]: tasks are reweighted based on uncertainty, (iii) **GradNorm** [10]: balance each task's gradient norm, (iv) **CosineSim** [13]: tasks are filtered out when having negative cosine similarity $\cos(\nabla \log p(\mathcal{T}_{a_k}|\theta), \nabla \log p(\mathcal{T}_m|\theta))$, (v) **OL_AUX** [31]: tasks have higher weights when the gradient inner product $\nabla \log p(\mathcal{T}_{a_k}|\theta)^T \nabla \log p(\mathcal{T}_m|\theta)$ is large. Besides, we also compare with grid search as an 'upper bound' of ARML. Since grid search is extremely expensive, we only compare with it when the task number is small (*e.g.* $K = 2$).

**Semi-supervised Learning (SSL)**   In SSL, one generally trains classifier with certain percentage of labeled data as the main task, and at the same time designs different losses on unlabeled data as auxiliary tasks. Specifically, we use *Self-supervised Semi-supervised Learning* (S4L) [55] as our baseline algorithm. S4L uses self-supervised methods on unlabeled part of training data, and trains classifier on labeled data as normal. Following [55], we use two kinds of self-supervised methods: *Rotation* and *Exemplar-MT*. In *Rotation*, we rotate each image by $[0°, 90°, 180°, 270°]$ and ask the network to predict the angle. In *Exemplar-MT*, the model is trained to extract feature invariant to a wide range of image transformations. Here we use random flipping, gaussian noise [8] and Cutout [12] as data augmentation. During training, each image is randomly augmented, and then features of original image and augmented image are encouraged to be close.

Table 1: Test error of semi-supervised learning on CIFAR-10 and SVHN. From top to bottom: purely-supervised method, state-of-the-art semi-supervised methods, and S4L with different reweighting schemes. * means multiple runs are needed.

|  | CIFAR-10 (4000 labels) | SVHN (1000 labels) |
|---|---|---|
| Supervised | $20.26 \pm .38$ | $12.83 \pm .47$ |
| Π-Model [28] | $16.37 \pm .63$ | $7.19 \pm .27$ |
| Mean Teacher [47] | $15.87 \pm .28$ | $5.65 \pm .47$ |
| VAT [41] | $13.86 \pm .27$ | $5.63 \pm .20$ |
| VAT + EntMin [17] | $\mathbf{13.13} \pm .39$ | $\mathbf{5.35} \pm .19$ |
| Pseudo-Label [29] | $17.78 \pm .57$ | $7.62 \pm .29$ |
| S4L (Uniform) | $15.67 \pm .29$ | $7.83 \pm .33$ |
| S4L + AdaLoss | $21.06 \pm .17$ | $11.53 \pm .39$ |
| S4L + GradNorm | $14.07 \pm .44$ | $7.68 \pm .13$ |
| S4L + CosineSim | $15.03 \pm .31$ | $7.02 \pm .25$ |
| S4L + OL_AUX | $16.07 \pm .51$ | $7.82 \pm .32$ |
| S4L + GridSearch* | $13.76 \pm .22$ | $6.07 \pm .17$ |
| S4L + ARML (ours) | $13.68 \pm .35$ | $5.89 \pm .22$ |

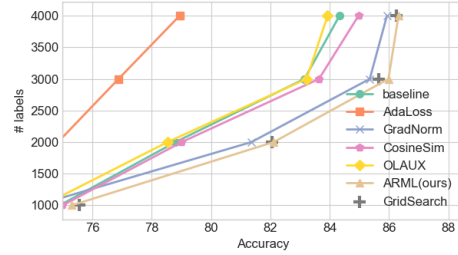

Figure 2: Amount of labeled data required to reach certain accuracy on CIFAR-10.

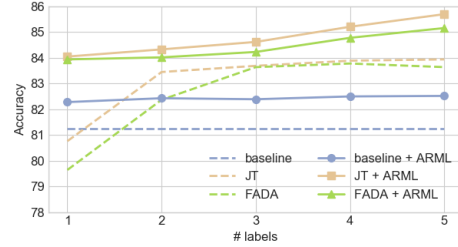

Figure 3: Accuracy of multi-source domain generalization with Art as target.

Based on S4L, we use task reweighting to adjust the weights for different self-supervised losses. Following the literature [41, 47], we test on two widely-used benchmarks: CIFAR-10 [27] with 4000 out of 45000 images labeled, and SVHN [40] with 1000 out of 65932 images labeled. We report test error of S4L with different reweighting schemes in Table 1, along with other SSL methods. We notice that, on both datasets, with the same amount of labeled data, ARML makes a better use of the data than uniform baseline as well as other reweighting methods. Remarkably, with only one pass, ARML is able to find the optimal weights while GridSearch needs multiple runs. S4L with our ARML applied is comparable to other state-of-the-art SSL methods. Notably, we only try *Rotation* and *Exemplar-MT*, while exploring more auxiliary tasks could further benefit the main task and we leave it for future study.

To see whether ARML can consistently reduce data requirement, we also test the amount of data required to reach different accuracy on CIFAR-10. As shown in Fig. 2, with ARML applied, we only need about half of labels to reach a decent performance. This also agrees with the results of GridSearch, showing the maximum improvement from auxiliary tasks during joint training.

Table 2: Test error of main task on CelebA.

|  | Test Error |
|---|---|
| Baseline | $6.70 \pm .18$ |
| AdaLoss [21] | $7.21 \pm .11$ |
| GradNorm [10] | $6.44 \pm .07$ |
| CosineSim [13] | $6.51 \pm .14$ |
| OL_AUX [31] | $6.32 \pm .17$ |
| ARML (ours) | $\mathbf{5.97 \pm .18}$ |

Table 3: Top 5 relative / irrelative attributes (auxiliary tasks) to the target attribute (main task) on CelebA.

| main task | most related tasks | least related tasks |
|---|---|---|
| 5_o_Clock_Shadow | Mustache<br>Bald<br>Sideburns<br>Rosy_Cheeks<br>Goatee | Mouth_Slightly_Open<br>Male<br>Attractive<br>Heavy_Makeup<br>Smiling |

**Multi-label Classification (MLC)** We also test our method in MLC. We use the CelebA dataset [34]. It contains 200K face images, each labeled with 40 binary attributes. We cast this into a MLC problem, where we randomly choose one target attribute as the main classification task, and other 39 as auxiliary tasks. To simulate our data-scarce setting, we only use 1% labels for main task.

We test different reweighting methods and list the results in Table 2. With the same amount of labeled data, ARML can help find better and more generalizable model parameters than baseline as well as other reweighting methods. This also implies that ARML has a consistent advantage even when handling a large number of tasks. For a further verification, we also check if the learned relationship between different face attributes is aligned with human's intuition. In Table 3, we list the top 5 auxiliary tasks with the highest weights, and also the top 5 with the lowest weights. As we can see, ARML has automatically picked attributes describing facial hair (*e.g.* Mustache, Sideburns, Goatee), which coincides with the main task 5_o_Clock_Shadow, another kind of facial hair. On the other hand, the tasks with low weights seem to be unrelated to the main task. This means ARML can actually learn the task relationship that matches our intuition.

## 3.2 ARML can Benefit Unsupervised Learning at Minimal Cost

In Sec. 3.1, we use ARML to reweight tasks and find a better prior for main task in order to compensate for data scarcity. Then one may naturally wonder whether this still works under situations where the main task has no labeled data at all (*e.g.* unsupervised learning). In fact, this is a meaningful question, not only because unsupervised learning is one of the most important problems in the community, but also because using auxiliary tasks is a mainstream of unsupervised learning methods [7, 16, 42]. Intuitively, as long as the family of prior $p_{\alpha}(\theta)$ is strong enough (which is determined by auxiliary tasks), we can always find a prior that gives a good model even without label information. However, if we want to use ARML to find the prior, at least *some* labeled data is required to estimate the gradient for main task (Eq. (A11)). Then the question becomes, how minimum of the data does ARML need to find a proper set of weights? More specifically, can we use as little data as possible (*e.g.* 1 or 5 labeled images per class) to make substantial improvement?

To answer the question, we conduct experiments in domain generalization, a well-studied unsupervised problem. In domain generalization, there is a target domain with no data (labeled or unlabeled), and multiple source domains with plenty of data. People usually train a model on source domains (auxiliary tasks) and transfer it to the target domain (main task). To use ARML, we relax the restriction a little by adding $N_m$ extra labeled images for target domain, where $N_m = 1, \cdots, 5$. This slightly relaxed setting is known as few-shot domain adaptation (FSDA) which was studied in [37], and we also add their FSDA results into comparison. For dataset selection, we use a common benchmark PACS [30] which contains four distinct domains of Photo, Art, Cartoon and Sketch. We pick each one as target domain and the other three as source domains which are reweighted by our ARML.

We first set $N_m = 5$ to see the results (Table 4). Here we include both state-of-the-art domain generalization methods [3, 7, 14] and FSDA methods [37]. Since they are orthogonal to ARML, we apply ARML on both types of methods to see the relative improvement. Let us first look at domain generalization methods. Here the baseline refers to training a model on source domains (auxiliary tasks) and directly testing on target domain (main task). If we use the extra 5 labels to reweight different source domains with ARML, we can make a non-trivial improvement, especially with Sketch as target (4% absolute improvement). Note that in "Baseline + ARML", we update $\theta$ using only classification loss on source data (auxiliary loss), and the extra labeled data in the target

Table 4: Results of multi-source domain generalization (w/ extra 5 labeled images per class in target domain). We list results with each of four domains as target domain. From top to down: domain generalization methods, FSDA methods and different methods equipped with ARML. JT is short for joint-training. † means the results we reproduced are higher than originally reported.

| Method | Extra label | Sketch | Art | Cartoon | Photo |
|---|---|---|---|---|---|
| Baseline[†] | ✗ | 75.34 | 81.25 | 77.35 | 95.93 |
| D-SAM [14] | ✗ | 77.83 | 77.33 | 72.43 | 95.30 |
| JiGen [7] | ✗ | 71.35 | 79.42 | 75.25 | 96.03 |
| Shape-bias [3] | ✗ | **78.62** | **83.01** | **79.39** | **96.83** |
| JT | ✓ | 78.52 | 83.94 | **81.36** | 97.01 |
| FADA [37] | ✓ | 79.23 | 83.64 | 79.39 | 97.07 |
| Baseline + ARML | ✓ | 79.35 | 82.52 | 77.30 | 95.99 |
| JT + ARML | ✓ | **80.47** | **85.70** | 81.01 | **97.22** |
| FADA + ARML | ✓ | 79.46 | 85.16 | 81.23 | 97.01 |

domain are just used for reweighting the auxiliary tasks, which means the improvement completely comes from task reweighting. Additionally, joint-training (JT) and FSDA methods also use extra labeled images by adding them into classification loss. If we further use the extra labels for task reweighting, then we can make a further improvement and reach a state-of-the-art performance.

We also test performance of ARML with $N_m = 1, \cdots, 5$. As an example, here we use Art as target domain. As shown in Fig. 3, ARML is able to improve the accuracy over different domain generalization or FSDA methods. Remarkably, when $N_m = 1$, although FSDA methods are under-performed, ARML can still bring an improvement of $\sim 4\%$ accuracy. This means ARML can benefit unsupervised domain generalization with as few as 1 labeled image per class.

### 3.3   ARML is Robust to Data Scarcity

Finally, we examine the robustness of our method. Due to the approximation made in Sec. 2.3, ARML may find a suboptimal solution. For example, in the true prior approximation (**P1**), we use $p(\mathcal{T}_m|\theta)$ to replace $p^*(\theta)$. When the size of $\mathcal{T}_m$ is large, these two should be close to each other. However, if we have less data, $p(\mathcal{T}_m|\theta)$ may also have high-value region outside $p^*(\theta)$ (*i.e.* 'overfitting' area), which may make the approximation inaccurate. To test the robustness of ARML, we check whether ARML can find similar task weights under different levels of data scarcity.

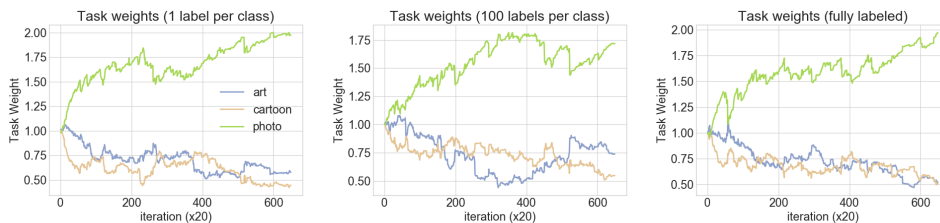

Figure 4: Change of task weights during training under different levels of data scarcity. From left to right: one-shot, partially labeled and fully labeled.

We conduct experiments on multi-source domain generalization with Art as target domain. We test three levels of data scarcity: few-shot (1 label per class), partly labeled (100 labels per class) and fully labeled ($\sim 300$ labels per class). We plot the change of task weights during training time in Fig. 4. We can see that task weights found by ARML are barely affected by data scarcity, even in few-shot scenario. This means ARML is able to find the optimal weights even with minimal guidance, verifying the rationality of approximation in Sec. 2.3 and the robustness of our method.

## 4 Related Work

**Additional Supervision from Auxiliary Tasks**    When there is not enough data to learn a task, it is common to introduce additional supervision from some related auxiliary tasks. For example, in semi-supervised learning, previous work has employed various kinds of manually-designed supervision on unlabeled data [41, 47, 55]. In reinforcement leaning, due to sample inefficiency, auxiliary tasks (*e.g.* vision prediction [36], reward prediction [46]) are jointly trained to speed up convergence. In transfer learning or domain adaptation, models are trained on related domains/tasks and generalize to unseen domains [3, 7, 48]. Learning using privileged information (LUPI) also employs additional knowledge (*e.g.* meta data, additional modality) during training time [20, 45, 50]. However, LUPI does not emphasize the scarcity of training data as in our problem setting.

**Multi-task Learning**    A highly related setting is multi-task learning (MTL). In MTL, models are trained to give high performance on different tasks simultaneously. Note that this is different from our setting because we only care about the performance on the main task. MTL is typically conducted through parameter sharing [2, 44], or prior sharing in a Bayesian manner [4, 19, 52, 53]. Parameter sharing and joint learning can achieve better generalization over learning each task independently [2], which also motivates our work. MTL has wide applications in areas including vision [6], language [11], speech [23], *etc*. We refer interested readers to this review [43].

**Adaptive Task Reweighting**    When learning multiple tasks, it is important to estimate the relationship between different tasks in order to balance multiple losses. In MTL, this is usually realized by task clustering through a mixture prior [4, 15, 35, 56]. However, this type of methods only screens out unrelated tasks without further differentiating related tasks. Another line of work balances multiple losses based on gradient norm [10] or uncertainty [21, 26]. In our problem setting, the focus is changed to estimate the relationship between the main task and auxiliary tasks. In [54] task relationship is estimated based on whether the representation learned for one task can be easily reused for another task, which requires exhaustive enumeration of all the tasks. In [1], the enumeration process is vastly simplified by only considering a local landscape in the parameter space. However, a local landscape may be insufficient to represent the whole parameter distribution, especially in high dimensional cases such as deep networks. Recently, algorithms have been designed to adaptively reweight multiple tasks on the fly. For example, in [13] tasks are filtered out when having opposite gradient direction to the main task. The most similar work to ours is [31], where the task relationship is also estimated from similarity between gradients. However, unlike our method, they use inner product as similarity metric with the goal of speeding up training.

## 5 Conclusion

In this work, we develop ARML, an algorithm to automatically reweight auxiliary tasks, so that the data requirement for the main task is minimized. We first formulate the weighted likelihood function of auxiliary tasks as a surrogate prior for the main task. Then the optimal weights are obtained by minimizing the divergence between the surrogate prior and the true prior. We design a practical algorithm by turning the optimization problem into minimizing the distance between main task gradient and auxiliary task gradients. We demonstrate its effectiveness and robustness in reducing the data requirement under various settings including the extreme case of only a few examples.

## Acknowledgments and Disclosure of Funding

Prof. Darrell's group was supported in part by DoD, BAIR and BDD. Prof. Saenko was supported by DARPA and NSF. Prof. Hoffman was supported by DARPA. The authors also acknowledge the valuable suggestions from Colorado Reed, Dinghuai Zhang, Qi Dai, and Ziqi Pang.

## Broader Impact

In this work we focus on solving the data scarcity problem of a main task using auxiliary tasks, and propose an algorithm to automatically reweight auxiliary tasks so that the data requirement on the main task is minimized. On the bright side, this could impact the industry and society from two

aspects. First, this may promote the landing of machine learning algorithms where labeled data is scarce or even unavailable, which is common in the real world. Second, our method can save the time and power resources wasted for manually tuning the auxiliary task weights with multiple runs, which is crucial in an era of environmental protection. However, our method may lead to negative consequences if it is not used right. For example, our method may be utilized to extract information from a private dataset or system with less data under the assistance of other auxiliary tasks. Besides, our method may still fail in some situations where the auxiliary tasks are strong regularization of the main task, which may not allow the use in applications where high precision and robustness are imperative.

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
