[Supplementary Material]

# Appendix to "Auxiliary Task Reweighting for Minimum-data Learning"

## 1 Additional Discussion on ARML

In this section we add more discussion on validity and soundness of ARML, especially on the three problems (**True Prior (P1)**, **Samples (P2)**, **Partition Function (P3)**), and how we resolve them (Sec. 3.3).

### 1.1 Full Version and Proof of Theorem 1 (P1)

In **True Prior (P1)** (Sec. 3.3) we use

$$\min_{\boldsymbol{\alpha}} E_{\theta \sim p^J} \log \frac{p^m(\theta)}{p_{\boldsymbol{\alpha}}(\theta)} \tag{A1}$$

as a surrogate objective for the original optimization problem

$$\min_{\boldsymbol{\alpha}} D_{\mathrm{KL}}(p^*(\theta) \parallel p_{\boldsymbol{\alpha}}(\theta)). \tag{A2}$$

In this section, we will first intuitively explain why optimizing (A1) can end up with a near-optimal solution for (A2), and what assumptions do we need to make. Then we will give the full version of Theorem 1 and also the proof.

Let $f(\boldsymbol{\alpha}) = E_{\theta \sim p^J} \log \frac{p^m(\theta)}{p_{\boldsymbol{\alpha}}(\theta)} = \frac{1}{Z(\boldsymbol{\alpha})} \int p^m(\theta) p_{\boldsymbol{\alpha}}(\theta) \log \frac{p^m(\theta)}{p_{\boldsymbol{\alpha}}(\theta)} d\theta$ be the optimization objective in (A1), where $p^J(\theta) = \frac{p^m(\theta) p_{\boldsymbol{\alpha}}(\theta)}{Z(\boldsymbol{\alpha})}$ and $Z(\boldsymbol{\alpha}) = \int p^m(\theta) p_{\boldsymbol{\alpha}}(\theta) d\theta$ is the normalization term. Assume $p^*(\theta)$ has a compact support set $S$. Then we can write $f(\boldsymbol{\alpha})$ as

$$
\begin{aligned}
f(\boldsymbol{\alpha}) &= \frac{1}{Z(\boldsymbol{\alpha})} \int_{\theta \in S} p^m(\theta) p_{\boldsymbol{\alpha}}(\theta) \log \frac{p^m(\theta)}{p_{\boldsymbol{\alpha}}(\theta)} d\theta + \frac{1}{Z(\boldsymbol{\alpha})} \int_{\theta \notin S} p^m(\theta) p_{\boldsymbol{\alpha}}(\theta) \log \frac{p^m(\theta)}{p_{\boldsymbol{\alpha}}(\theta)} d\theta \\
&= \frac{Z(S; \boldsymbol{\alpha})}{Z(S; \boldsymbol{\alpha}) + Z(\bar{S}; \boldsymbol{\alpha})} \int_{\theta \in S} \frac{p^m(\theta) p_{\boldsymbol{\alpha}}(\theta)}{Z(S; \boldsymbol{\alpha})} \log \frac{p^m(\theta)}{p_{\boldsymbol{\alpha}}(\theta)} d\theta \\
&\quad + \frac{Z(\bar{S}; \boldsymbol{\alpha})}{Z(S; \boldsymbol{\alpha}) + Z(\bar{S}; \boldsymbol{\alpha})} \int_{\theta \notin S} \frac{p^m(\theta) p_{\boldsymbol{\alpha}}(\theta)}{Z(\bar{S}; \boldsymbol{\alpha})} \log \frac{p^m(\theta)}{p_{\boldsymbol{\alpha}}(\theta)} d\theta \\
&= f(\boldsymbol{\alpha}; S) + f(\boldsymbol{\alpha}; \bar{S}),
\end{aligned}
\tag{A3}
$$

where we denote the first and second term by $f(\boldsymbol{\alpha}; S)$ and $f(\boldsymbol{\alpha}; \bar{S})$ respectively, $Z(S; \boldsymbol{\alpha}) = \int_{\theta \in S} p^m(\theta) p_{\boldsymbol{\alpha}}(\theta) d\theta$ and $Z(\bar{S}; \boldsymbol{\alpha}) = \int_{\theta \notin S} p^m(\theta) p_{\boldsymbol{\alpha}}(\theta) d\theta$ are the normalization terms inside and outside $S$.

To build the connection between the surrogate objective $f(\boldsymbol{\alpha})$ and the original objective $KL_{\boldsymbol{\alpha}} := D_{\mathrm{KL}}(p^*(\theta) \parallel p_{\boldsymbol{\alpha}}(\theta))$, we make the following assumption,

**Assumption 1.** *The support set $S$ is small so that $p_{\boldsymbol{\alpha}}(\theta)$ and $p^m(\theta)$ are constants inside $S$, and $p^*(\theta)$ is uniform in $S$.*

21  This assumption is reasonable when $S$ is really informative, which we assume is the case for the true
22  prior $p^*(\theta)$ [3]. With this assumption, we have

$$KL_{\boldsymbol{\alpha}} = \int_{\theta \in S} p^*(\theta) \log \frac{p^*(\theta)}{p_{\boldsymbol{\alpha}}(\theta)} d\theta = \log \frac{p^*(\theta^*)}{p_{\boldsymbol{\alpha}}(\theta^*)} \cdot \int_{\theta \in S} p^*(\theta) d\theta = \log \frac{p^*(\theta^*)}{p_{\boldsymbol{\alpha}}(\theta^*)}, \tag{A4}$$

23  where $\theta^* \in S$ is the optimal parameter. We can also write $f(\boldsymbol{\alpha}; S)$ as

$$
\begin{aligned}
f(\boldsymbol{\alpha}; S) &= \frac{Z(S; \boldsymbol{\alpha})}{Z(S; \boldsymbol{\alpha}) + Z(\bar{S}; \boldsymbol{\alpha})} \int_{\theta \in S} \frac{p^m(\theta) p_{\boldsymbol{\alpha}}(\theta)}{Z(S; \boldsymbol{\alpha})} \log \frac{p^m(\theta)}{p_{\boldsymbol{\alpha}}(\theta)} d\theta \\
&= \frac{Z(S; \boldsymbol{\alpha})}{Z(S; \boldsymbol{\alpha}) + Z(\bar{S}; \boldsymbol{\alpha})} \log \frac{p^m(\theta^*)}{p_{\boldsymbol{\alpha}}(\theta^*)} \cdot \int_{\theta \in S} \frac{p^m(\theta) p_{\boldsymbol{\alpha}}(\theta)}{Z(S; \boldsymbol{\alpha})} d\theta \\
&= \frac{Z(S; \boldsymbol{\alpha})}{Z(S; \boldsymbol{\alpha}) + Z(\bar{S}; \boldsymbol{\alpha})} \log \frac{p^m(\theta^*)}{p_{\boldsymbol{\alpha}}(\theta^*)} \\
&= \frac{Z(S; \boldsymbol{\alpha})}{Z(S; \boldsymbol{\alpha}) + Z(\bar{S}; \boldsymbol{\alpha})} (\log \frac{p^*(\theta^*)}{p_{\boldsymbol{\alpha}}(\theta^*)} + \log \frac{p^m(\theta^*)}{p^*(\theta^*)}) \\
&= \frac{Z(S; \boldsymbol{\alpha})}{Z(S; \boldsymbol{\alpha}) + Z(\bar{S}; \boldsymbol{\alpha})} (KL_{\boldsymbol{\alpha}} + C_1),
\end{aligned} \tag{A5}
$$

24  where $C_1 = \log \frac{p^m(\theta^*)}{p^*(\theta^*)}$ is a constant invariant to $\boldsymbol{\alpha}$. Since $p^m(\theta)$ also covers other "overfitting" area
25  other than $S$, we can assume that $p^*(\theta^*) \geq p^m(\theta^*)$, which gives $C_1 \leq 0$. Furthermore, we can notice
26  that

$$Z(S; \boldsymbol{\alpha}) = \int_{\theta \in S} p^m(\theta) p_{\boldsymbol{\alpha}}(\theta) d\theta = \int_{\theta \in S} \frac{p^m(\theta) p_{\boldsymbol{\alpha}}(\theta)}{p^*(\theta)} p^*(\theta) d\theta = \frac{p^m(\theta^*) p_{\boldsymbol{\alpha}}(\theta^*)}{p^*(\theta^*)} = C_2 e^{-KL_{\boldsymbol{\alpha}}}, \tag{A6}$$

27  where $C_2 = p^m(\theta^*)$ is a constant invariant to $\boldsymbol{\alpha}$. Then we can write $f(\boldsymbol{\alpha}; S)$ as

$$f(\boldsymbol{\alpha}; S) = \frac{C_2 e^{-KL_{\boldsymbol{\alpha}}}}{C_2 e^{-KL_{\boldsymbol{\alpha}}} + Z(\bar{S}; \boldsymbol{\alpha})} (KL_{\boldsymbol{\alpha}} + C_1). \tag{A7}$$

28  In this way, we build the connection between the surrogate objective $f(\boldsymbol{\alpha})$ and the original objective
29  $KL_{\boldsymbol{\alpha}}$.

30  Now we give an intuitive explanation for why optimizing $f(\boldsymbol{\alpha})$ gives a small $KL_{\boldsymbol{\alpha}}$ as well. We can
31  write $f(\boldsymbol{\alpha})$ as

$$
\begin{aligned}
f(\boldsymbol{\alpha}) &= f(\boldsymbol{\alpha}; S) + f(\boldsymbol{\alpha}; \bar{S}) \\
&= \frac{C_2 e^{-KL_{\boldsymbol{\alpha}}}}{C_2 e^{-KL_{\boldsymbol{\alpha}}} + Z(\bar{S}; \boldsymbol{\alpha})} (KL_{\boldsymbol{\alpha}} + C_1) + \frac{Z(\bar{S}; \boldsymbol{\alpha})}{C_2 e^{-KL_{\boldsymbol{\alpha}}} + Z(\bar{S}; \boldsymbol{\alpha})} \int_{\theta \in \bar{S}} \frac{p^m(\theta) p_{\boldsymbol{\alpha}}(\theta)}{Z(\bar{S}; \boldsymbol{\alpha})} \log \frac{p^m(\theta)}{p_{\boldsymbol{\alpha}}(\theta)} d\theta.
\end{aligned} \tag{A8}
$$

32  As one can notice, $f(\boldsymbol{\alpha})$ not only depends on $KL_{\boldsymbol{\alpha}}$, but also on $Z(\bar{S}; \boldsymbol{\alpha})$ and the integral
33  $\int_{\theta \in \bar{S}} \frac{p^m(\theta) p_{\boldsymbol{\alpha}}(\theta)}{Z(\bar{S}; \boldsymbol{\alpha})} \log \frac{p^m(\theta)}{p_{\boldsymbol{\alpha}}(\theta)} d\theta$. First we remove the dependency on the integral by taking its lower
34  bound and upper bound. Concretely, with Jensen's inequality, we have

$$\int_{\theta \in \bar{S}} \frac{p^m(\theta) p_{\boldsymbol{\alpha}}(\theta)}{Z(\bar{S}; \boldsymbol{\alpha})} \log \frac{p^m(\theta)}{p_{\boldsymbol{\alpha}}(\theta)} d\theta \leq \log \frac{\int_{\theta \in \bar{S}} (p^m(\theta))^2 d\theta}{Z(\bar{S}; \boldsymbol{\alpha})} = \log \frac{C_3}{Z(\bar{S}; \boldsymbol{\alpha})}, \tag{A9}$$

35  where $C_3 = \int_{\theta \in \bar{S}} (p^m(\theta))^2 d\theta$ is a constant invariant to $\boldsymbol{\alpha}$. Likewise, we have

$$
\begin{aligned}
\int_{\theta \in \bar{S}} \frac{p^m(\theta) p_{\boldsymbol{\alpha}}(\theta)}{Z(\bar{S}; \boldsymbol{\alpha})} \log \frac{p^m(\theta)}{p_{\boldsymbol{\alpha}}(\theta)} d\theta &= \int_{\theta \in \bar{S}} -\frac{p^m(\theta) p_{\boldsymbol{\alpha}}(\theta)}{Z(\bar{S}; \boldsymbol{\alpha})} \log \frac{p_{\boldsymbol{\alpha}}(\theta)}{p^m(\theta)} d\theta \\
&\geq -\log \frac{\int_{\theta \in \bar{S}} (p_{\boldsymbol{\alpha}}(\theta))^2 d\theta}{Z(\bar{S}; \boldsymbol{\alpha})} \\
&\geq -\log \frac{C_4}{Z(\bar{S}; \boldsymbol{\alpha})},
\end{aligned} \tag{A10}
$$

36  where $C_4 = \max_{\boldsymbol{\alpha}} \int_{\theta \in \bar{S}} (p_{\boldsymbol{\alpha}}(\theta))^2 d\theta$ is a constant invariant to $\boldsymbol{\alpha}$. In this way, we get the lower bound
37  and upper bound for $f(\boldsymbol{\alpha})$:

$$
\begin{aligned}
f(\boldsymbol{\alpha}) &\geq f_l(\boldsymbol{\alpha}) = \frac{C_2 e^{-KL_{\boldsymbol{\alpha}}}}{C_2 e^{-KL_{\boldsymbol{\alpha}}} + Z(\bar{S}; \boldsymbol{\alpha})} (KL_{\boldsymbol{\alpha}} + C_1) - \frac{Z(\bar{S}; \boldsymbol{\alpha})}{C_2 e^{-KL_{\boldsymbol{\alpha}}} + Z(\bar{S}; \boldsymbol{\alpha})} \log \frac{C_4}{Z(\bar{S}; \boldsymbol{\alpha})}, \\
f(\boldsymbol{\alpha}) &\leq f_u(\boldsymbol{\alpha}) = \frac{C_2 e^{-KL_{\boldsymbol{\alpha}}}}{C_2 e^{-KL_{\boldsymbol{\alpha}}} + Z(\bar{S}; \boldsymbol{\alpha})} (KL_{\boldsymbol{\alpha}} + C_1) + \frac{Z(\bar{S}; \boldsymbol{\alpha})}{C_2 e^{-KL_{\boldsymbol{\alpha}}} + Z(\bar{S}; \boldsymbol{\alpha})} \log \frac{C_3}{Z(\bar{S}; \boldsymbol{\alpha})}.
\end{aligned} \tag{A11}
$$

(a) $e^{-KL_{\alpha^*}}$ is large.  (b) $e^{-KL_{\alpha^*}}$ is small.

Figure 1: $f(e^{-KL_{\alpha}})$'s upper bound $f_u(e^{-KL_{\alpha}})$ (golden line) and lower bound $f_l(e^{-KL_{\alpha}})$ (blue line). $\boldsymbol{\alpha}^* = \arg\max_{\boldsymbol{\alpha}}(e^{-KL_{\alpha}}) = \arg\min_{\boldsymbol{\alpha}} KL_{\boldsymbol{\alpha}}$ denotes the largest $e^{-KL_{\alpha}}$ we could possibly reach. Shaded region denotes where $(e^{-KL_{\hat{\alpha}}}, f(e^{-KL_{\hat{\alpha}}}))$ could possibly be.

We plot $f_l$ and $f_u$ as functions of $e^{-KL_{\alpha}}$ in Fig. 1 (here we assume $Z(\bar{S}; \boldsymbol{\alpha})$ is constant w.r.t. $\boldsymbol{\alpha}$ for brevity). $f(\boldsymbol{\alpha})$ lies between the upper bound (golden line) and the lower bound (blue line).

Our goal is to find the optimal $\boldsymbol{\alpha}^*$ that minimizes $KL_{\boldsymbol{\alpha}}$, *i.e.*, $\boldsymbol{\alpha}^* = \arg\min_{\boldsymbol{\alpha}} KL_{\boldsymbol{\alpha}} = \arg\max_{\boldsymbol{\alpha}} e^{-KL_{\alpha}}$. By optimizing $f(\boldsymbol{\alpha})$, we end up with a suboptimal $\hat{\alpha} = \arg\min_{\boldsymbol{\alpha}} f(\boldsymbol{\alpha})$. Ideally, we hope that $KL_{\hat{\alpha}}$ is close to $KL_{\boldsymbol{\alpha}^*}$, which means when we minimize $f(\hat{\alpha})$, we also get a large $e^{-KL_{\hat{\alpha}}}$. This is the case when $e^{-KL_{\alpha^*}}$ is large (see Fig. 1a). When $e^{-KL_{\alpha^*}}$ is large, the upper bound $f_u$ and the lower bound $f_l$ are close to each other around $e^{-KL_{\alpha^*}}$ (this is the case when $Z(\bar{S}; \boldsymbol{\alpha})$ is small). Since we have

$$f_l(e^{-KL_{\hat{\alpha}}}) \leq f(e^{-KL_{\hat{\alpha}}}) \leq f(e^{-KL_{\alpha^*}}) \leq f_u(e^{-KL_{\alpha^*}}), \tag{A12}$$

we can assert that $(e^{-KL_{\hat{\alpha}}}, f(e^{-KL_{\hat{\alpha}}}))$ lies in the shaded region, because if $e^{-KL_{\hat{\alpha}}}$ is on the left side of the region, we have $f(e^{-KL_{\hat{\alpha}}}) \geq f_u(e^{-KL_{\alpha^*}})$ which is contradictary to (A12), and if $e^{-KL_{\hat{\alpha}}}$ cannot be on the right side of the region because $e^{-KL_{\alpha^*}}$ is the furthest we can go. Since the shaded region is small, $KL_{\hat{\alpha}}$ is thus close to the optimal solution $KL_{\boldsymbol{\alpha}^*}$.

Unfortunately, this may not hold anymore when $e^{-KL_{\alpha^*}}$ is small (see Fig. 1b). This is because $f_l$ will reach a local minima when $e^{-KL_{\alpha}} \to 0$. If $e^{-KL_{\alpha^*}}$ is not large enough, it may be higher than $\lim_{e^{-KL_{\alpha}} \to 0} f_l(e^{-KL_{\alpha}})$, which means the shaded region near y-axis is also included. In this region $f(\boldsymbol{\alpha})$ could be really small (which is the goal when optimizing the surrogate objective $f(\boldsymbol{\alpha})$), but $KL_{\boldsymbol{\alpha}}$ could be extremely large.

To avoid this situation, we only have to assume that

$$f_u(e^{-KL_{\alpha^*}}) \leq \lim_{e^{-KL_{\alpha}} \to 0} f_l(e^{-KL_{\alpha}}) = -\log \frac{C_4}{Z(\bar{S}; \boldsymbol{\alpha})}, \tag{A13}$$

or if we denote $\gamma_1 = \min_{\boldsymbol{\alpha}} Z(\bar{S}; \boldsymbol{\alpha})$ and $\gamma_2 = \max_{\boldsymbol{\alpha}} Z(\bar{S}; \boldsymbol{\alpha})$, then we only need the following assumption:

**Assumption 2.** *The optimal $KL_{\boldsymbol{\alpha}^*}$ is small so that $f_u(e^{-KL_{\alpha^*}}) \leq -\log \frac{C_4}{\gamma_1}$.*

This assumption holds as long as there is at least one task that is related to the main task (having a small $KL_{\boldsymbol{\alpha}}$), which is reasonable because if all the tasks are unrelated, then reweighing is also meaningless. See the remark below for more discussion on the validity of the assumption.

Now we give the formal version of the theorem:

**Theorem 1.** *(formal version) With Assumption 1, 2, if $\gamma_2 \leq \min(\frac{C_3}{e}, \frac{C_4}{e})$, then we have*

$$KL_{\hat{\alpha}} \leq KL_{\boldsymbol{\alpha}^*} + \frac{2\gamma_2^2}{C} \log \frac{C'}{\gamma_2} \tag{A14}$$

*Proof.* From Assumption 2 we have

$$\frac{C_2 e^{-KL_{\alpha^*}}}{C_2 e^{-KL_{\alpha^*}} + Z(\bar{S}; \boldsymbol{\alpha}^*)}(KL_{\boldsymbol{\alpha}^*} + C_1) + \frac{Z(\bar{S}; \boldsymbol{\alpha}^*)}{C_2 e^{-KL_{\alpha^*}} + Z(\bar{S}; \boldsymbol{\alpha}^*)} \log \frac{C_3}{Z(\bar{S}; \boldsymbol{\alpha}^*)} \leq -\log \frac{C_4}{\gamma_1}. \tag{A15}$$

Since $\gamma_2 \leq C_3$ and $\gamma_2 \leq C_4$, we have $\log \frac{C_3}{Z(\bar{S};\boldsymbol{\alpha}^*)} \geq \log \frac{C_3}{\gamma_2} \geq 0$, and $-\log \frac{C_4}{\gamma_1} \leq -\log \frac{C_4}{\gamma_2} \leq 0$. Then leaves us $KL_{\boldsymbol{\alpha}^*} + C_1 \leq 0$ in order to make (A15) satisfied. Then we can relax (A15) into

$$KL_{\boldsymbol{\alpha}^*} + C_1 \leq -\log \frac{C_4}{\gamma_1}, \tag{A16}$$

which gives

$$C_2 e^{-KL_{\boldsymbol{\alpha}^*}} \geq \frac{C_5}{\gamma_1}, \tag{A17}$$

where $C_5 = C_4 C_2 e^{C_1}$. This bounds the value of $KL_{\boldsymbol{\alpha}^*}$.

Moreover, from (A12) and Assumption 2 we have

$$f_l(e^{-KL_{\hat{\boldsymbol{\alpha}}}}) \leq f_u(e^{-KL_{\boldsymbol{\alpha}^*}}) \leq -\log \frac{C_4}{\gamma_1}, \tag{A18}$$

which gives

$$f_l(e^{-KL_{\hat{\boldsymbol{\alpha}}}}) = \frac{C_2 e^{-KL_{\hat{\boldsymbol{\alpha}}}}}{C_2 e^{-KL_{\hat{\boldsymbol{\alpha}}}} + Z(\bar{S};\hat{\boldsymbol{\alpha}})}(KL_{\hat{\boldsymbol{\alpha}}} + C_1) - \frac{Z(\bar{S};\hat{\boldsymbol{\alpha}})}{C_2 e^{-KL_{\hat{\boldsymbol{\alpha}}}} + Z(\bar{S};\hat{\boldsymbol{\alpha}})} \log \frac{C_4}{Z(\bar{S};\hat{\boldsymbol{\alpha}})} \leq -\log \frac{C_4}{\gamma_1}. \tag{A19}$$

Since $Z(\bar{S};\hat{\boldsymbol{\alpha}}) \geq \gamma_1$, we can relax (A19) into

$$\frac{C_2 e^{-KL_{\hat{\boldsymbol{\alpha}}}}}{C_2 e^{-KL_{\hat{\boldsymbol{\alpha}}}} + Z(\bar{S};\hat{\boldsymbol{\alpha}})}(KL_{\hat{\boldsymbol{\alpha}}} + C_1) - \frac{Z(\bar{S};\hat{\boldsymbol{\alpha}})}{C_2 e^{-KL_{\hat{\boldsymbol{\alpha}}}} + Z(\bar{S};\hat{\boldsymbol{\alpha}})} \log \frac{C_4}{Z(\bar{S};\hat{\boldsymbol{\alpha}})} \leq -\log \frac{C_4}{Z(\bar{S};\hat{\boldsymbol{\alpha}})}, \tag{A20}$$

which can be simplified into

$$KL_{\hat{\boldsymbol{\alpha}}} + C_1 \leq -\log \frac{C_4}{Z(\bar{S};\hat{\boldsymbol{\alpha}})} \leq -\log \frac{C_4}{\gamma_2}, \tag{A21}$$

which means

$$C_2 e^{-KL_{\hat{\boldsymbol{\alpha}}}} \geq \frac{C_5}{\gamma_2}. \tag{A22}$$

This bounds the value of $KL_{\hat{\boldsymbol{\alpha}}}$.

Now we build the connection between $KL_{\hat{\boldsymbol{\alpha}}}$ and $KL_{\boldsymbol{\alpha}^*}$. Since $f_l(e^{-KL_{\hat{\boldsymbol{\alpha}}}}) \leq f_u(e^{-KL_{\boldsymbol{\alpha}^*}})$, we have

$$\frac{C_2 e^{-KL_{\hat{\boldsymbol{\alpha}}}}}{C_2 e^{-KL_{\hat{\boldsymbol{\alpha}}}} + Z(\bar{S};\hat{\boldsymbol{\alpha}})}(KL_{\hat{\boldsymbol{\alpha}}} + C_1) - \frac{Z(\bar{S};\hat{\boldsymbol{\alpha}})}{C_2 e^{-KL_{\hat{\boldsymbol{\alpha}}}} + Z(\bar{S};\hat{\boldsymbol{\alpha}})} \log \frac{C_4}{Z(\bar{S};\hat{\boldsymbol{\alpha}})}$$
$$\leq \frac{C_2 e^{-KL_{\boldsymbol{\alpha}^*}}}{C_2 e^{-KL_{\boldsymbol{\alpha}^*}} + Z(\bar{S};\boldsymbol{\alpha}^*)}(KL_{\boldsymbol{\alpha}^*} + C_1) + \frac{Z(\bar{S};\boldsymbol{\alpha}^*)}{C_2 e^{-KL_{\boldsymbol{\alpha}^*}} + Z(\bar{S};\boldsymbol{\alpha}^*)} \log \frac{C_3}{Z(\bar{S};\boldsymbol{\alpha}^*)}. \tag{A23}$$

Since $KL_{\hat{\boldsymbol{\alpha}}} + C_1 \leq -\log \frac{C_4}{\gamma_2} \leq 0$, $KL_{\boldsymbol{\alpha}^*} \geq 0$, and also with (A17) and (A22), we can relax (A23) into

$$KL_{\hat{\boldsymbol{\alpha}}} + C_1 - \frac{Z(\bar{S};\hat{\boldsymbol{\alpha}})}{C_5/\gamma_2} \log \frac{C_4}{Z(\bar{S};\hat{\boldsymbol{\alpha}})}$$
$$\leq KL_{\boldsymbol{\alpha}^*} + \frac{C_2 e^{-KL_{\boldsymbol{\alpha}^*}}}{C_2 e^{-KL_{\boldsymbol{\alpha}^*}} + Z(\bar{S};\boldsymbol{\alpha}^*)}C_1 + \frac{Z(\bar{S};\boldsymbol{\alpha}^*)}{C_5/\gamma_1} \log \frac{C_3}{Z(\bar{S};\boldsymbol{\alpha}^*)}, \tag{A24}$$

which gives

$$KL_{\hat{\boldsymbol{\alpha}}} \leq KL_{\boldsymbol{\alpha}^*} - \frac{Z(\bar{S};\boldsymbol{\alpha}^*)}{C_2 e^{-KL_{\boldsymbol{\alpha}^*}} + Z(\bar{S};\boldsymbol{\alpha}^*)}C_1 + \frac{Z(\bar{S};\hat{\boldsymbol{\alpha}})}{C_5/\gamma_2} \log \frac{C_4}{Z(\bar{S};\hat{\boldsymbol{\alpha}})} + \frac{Z(\bar{S};\boldsymbol{\alpha}^*)}{C_5/\gamma_1} \log \frac{C_3}{Z(\bar{S};\boldsymbol{\alpha}^*)}. \tag{A25}$$

Since $Z(\bar{S};\hat{\boldsymbol{\alpha}}) \leq \gamma_2 \leq \frac{C_4}{e}$, we have $Z(\bar{S};\hat{\boldsymbol{\alpha}}) \log \frac{C_4}{Z(\bar{S};\hat{\boldsymbol{\alpha}})} \leq \gamma_2 \log \frac{C_4}{\gamma_2}$. Similarly, we have $Z(\bar{S};\boldsymbol{\alpha}^*) \log \frac{C_3}{Z(\bar{S};\boldsymbol{\alpha}^*)} \leq \gamma_2 \log \frac{C_3}{\gamma_2}$. Then we have

$$KL_{\hat{\boldsymbol{\alpha}}} \leq KL_{\boldsymbol{\alpha}^*} - \frac{Z(\bar{S};\boldsymbol{\alpha}^*)}{C_2 e^{-KL_{\boldsymbol{\alpha}^*}} + Z(\bar{S};\boldsymbol{\alpha}^*)}C_1 + \frac{\gamma_2^2}{C_5} \log \frac{C_4}{\gamma_2} + \frac{\gamma_2^2}{C_5} \log \frac{C_3}{\gamma_2}. \tag{A26}$$

Since $C_1 \leq 0$, we can get

$$KL_{\hat{\boldsymbol{\alpha}}} \leq KL_{\boldsymbol{\alpha}^*} + \frac{\gamma_2^2}{C_5}(-C_1) + \frac{\gamma_2^2}{C_5} \log \frac{C_4}{\gamma_2} + \frac{\gamma_2^2}{C_5} \log \frac{C_3}{\gamma_2}, \tag{A27}$$

which gives

$$KL_{\hat{\boldsymbol{\alpha}}} \leq KL_{\boldsymbol{\alpha}^*} + \frac{2\gamma_2^2}{C_5} \log \frac{C_6}{\gamma_2}, \tag{A28}$$

where $C_6 = \sqrt{C_3 C_4 e^{-C_1}}$.

$\square$

**Remark.** *From Theorem 1 we see that $KL_{\hat{\boldsymbol{\alpha}}}$ is close to $KL_{\boldsymbol{\alpha}^*}$ as long as $\gamma_2$ is small. One may notice that $\gamma_2$ cannot be arbitrarily small because from (A22) we have*

$$\frac{C_5}{\gamma_2} \leq C_2 e^{-KL_{\hat{\boldsymbol{\alpha}}}} \leq C_2, \tag{A29}$$

*which means*

$$\gamma_2 \geq \frac{C_5}{C_2} = C_4 e^{C_1}. \tag{A30}$$

*However, we can safely assume that*

$$C_1 = \log \frac{p^m(\theta^*)}{p^*(\theta^*)} \ll 0 \tag{A31}$$

*since $p^*$ is much more informative than $p^m$, especially when labeled data for the main task is scarce. This means $\gamma_2$ can be extremely small as long as $C_1$ is small, which makes $KL_{\hat{\boldsymbol{\alpha}}}$ close to $KL_{\boldsymbol{\alpha}^*}$. Similarly, Assumption 2 can easily hold as long as $C_1$ is small.*

## 1.2 Sampling through Langevin Dynamics (P2)

In **Samples (P2)** we use Langevin dynamics [16, 22] to sample from the distribution $p^J$. Concretely, at each iteration, we update $\theta$ by

$$\theta_{t+1} = \theta_t - \epsilon_t \nabla \mathcal{L}(\theta_t) + \eta_t, \tag{A32}$$

where $\mathcal{L}(\theta) \propto -\log p^J(\theta)$ is the joint loss, and $\eta_t \sim N(0, 2\epsilon_t)$ is a Gaussian noise. In this way, $\theta_t$ converges to samples from $p^J$, which can be used to estimate our optimization objective. However, since we normally use a mini-batch estimator $\hat{\mathcal{L}}(\theta)$ to approximate $\mathcal{L}(\theta)$, this may introduce additional noise other than $\eta_t$, which may make the sampling procedure inaccurate. In [22] it is proposed to anneal the learning rate to zero so that the gradient stochasticity is dominated by the injected noise, thus alleviating the impact of mini-batch estimator. However we find in practice that the gradient noise is negligible compared to the injected noise (Table 1). Therefore, we ignore the gradient noise and directly inject the noise $\eta_t$ into the updating step.

Table 1: Standard deviation of different types of noise. We find that the gradient noise is negligible compared to the injected noise.

|  | Standard deviation |
| --- | --- |
| Gradient Noise | $\sim 10^{-6}$ |
| Injected Noise | $\sim 10^{-3}$ |

## 1.3 Score Function and Fisher Divergence (P3)

In **Partition Function (P3)** we propose to minimize

$$\min_{\boldsymbol{\alpha}} E_{\theta \sim p^J} \| \nabla \log p(\mathcal{T}_m | \theta) - \nabla \log p_{\boldsymbol{\alpha}}(\theta) \|_2^2 \tag{A33}$$

as our final objective. Notice that

$$\begin{aligned}
&\min_{\boldsymbol{\alpha}} E_{\theta \sim p^J} \| \nabla \log p(\mathcal{T}_m | \theta) - \nabla \log p_{\boldsymbol{\alpha}}(\theta) \|_2^2 \\
\Leftrightarrow\ &\min_{\boldsymbol{\alpha}} E_{\theta \sim p^J} \| \nabla \log p^m(\theta) - \nabla \log p_{\boldsymbol{\alpha}}(\theta) \|_2^2 \\
\Leftrightarrow\ &\min_{\boldsymbol{\alpha}} E_{\theta \sim p^J} \| \nabla \log(p^m(\theta) \cdot p_{\boldsymbol{\alpha}}(\theta)) - 2 \cdot \nabla \log p_{\boldsymbol{\alpha}}(\theta) \|_2^2 \\
\Leftrightarrow\ &\min_{\boldsymbol{\alpha}} E_{\theta \sim p^J} \| \nabla \log p^J(\theta) - \nabla \log p_{\boldsymbol{\alpha}}^2(\theta) \|_2^2 \\
\Leftrightarrow\ &\min_{\boldsymbol{\alpha}} F(p^J(\theta) \| \frac{1}{Z'(\boldsymbol{\alpha})} p_{\boldsymbol{\alpha}}^2(\theta)),
\end{aligned} \tag{A34}$$

where $F(p(\theta) \parallel q(\theta)) = E_{\theta \sim p} \|\nabla \log p(\theta) - \nabla \log q(\theta)\|_2^2$ is the *Fisher divergence*, and $Z'(\boldsymbol{\alpha}) = \int p_{\boldsymbol{\alpha}}^2(\theta) d\theta$ is the normalization term. This means, by optimizing (A33), we are actually minimizing the Fisher divergence between $p^J(\theta)$ and $\frac{1}{Z'(\boldsymbol{\alpha})} p_{\boldsymbol{\alpha}}^2(\theta)$. As pointed by [8, 13], Fisher divergence is stronger than KL divergence, which means by minimizing $F(p^J(\theta) \parallel \frac{1}{Z'(\boldsymbol{\alpha})} p_{\boldsymbol{\alpha}}^2(\theta))$, the KL divergence $D_{KL}(p^J(\theta) \parallel \frac{1}{Z'(\boldsymbol{\alpha})} p_{\boldsymbol{\alpha}}^2(\theta))$ is also bounded near the optimum up to a small error.

Therefore, optimizing (A33) is equivalent to minimizing $D_{KL}(p^J(\theta) \parallel \frac{1}{Z'(\boldsymbol{\alpha})} p_{\boldsymbol{\alpha}}^2(\theta))$. Notice that

$$
\begin{aligned}
&\min_{\boldsymbol{\alpha}} D_{KL}(p^J(\theta) \parallel \frac{1}{Z'(\boldsymbol{\alpha})} p_{\boldsymbol{\alpha}}^2(\theta)) \\
\Leftrightarrow\ &\min_{\boldsymbol{\alpha}} \int p^J(\theta) \log \frac{p^J(\theta)}{\frac{1}{Z'(\boldsymbol{\alpha})} p_{\boldsymbol{\alpha}}^2(\theta)} d\theta \\
\Leftrightarrow\ &\min_{\boldsymbol{\alpha}} \int p^J(\theta) \log \frac{\frac{1}{Z(\boldsymbol{\alpha})} p^m(\theta) p_{\boldsymbol{\alpha}}(\theta)}{\frac{1}{Z'(\boldsymbol{\alpha})} p_{\boldsymbol{\alpha}}^2(\theta)} d\theta \\
\Leftrightarrow\ &\min_{\boldsymbol{\alpha}} \int p^J(\theta) \log \frac{p^m(\theta)}{p_{\boldsymbol{\alpha}}(\theta)} d\theta + \log \frac{Z'(\boldsymbol{\alpha})}{Z(\boldsymbol{\alpha})} \\
\Leftrightarrow\ &\min_{\boldsymbol{\alpha}} \int p^J(\theta) \log \frac{p^m(\theta)}{p_{\boldsymbol{\alpha}}(\theta)} d\theta + \log \frac{\int p_{\boldsymbol{\alpha}}^2(\theta) d\theta}{\int p^m(\theta) p_{\boldsymbol{\alpha}}(\theta) d\theta}
\end{aligned}
\tag{A35}
$$

is different from (A1) only on the $\log \frac{\int p_{\boldsymbol{\alpha}}^2(\theta) d\theta}{\int p^m(\theta) p_{\boldsymbol{\alpha}}(\theta) d\theta}$ term. To analyze the impact of this additional term, we assume that the likelihood function of each auxiliary task is a Gaussian, *i.e.*, $p(\mathcal{T}_{a_k}|\theta) \propto N(\theta|\theta_k, \boldsymbol{\Sigma})$, with mean $\theta_k$ and covariance $\boldsymbol{\Sigma}$. Then we have $p_{\boldsymbol{\alpha}}(\theta) = N(\theta| \sum_k \alpha_k \theta_k / K, \boldsymbol{\Sigma}/K)$ (note that $\sum_k \alpha_k = K$). In this case $\int p_{\boldsymbol{\alpha}}^2(\theta) d\theta$ only depends on $\boldsymbol{\Sigma}$ and is invariant to $\boldsymbol{\alpha}$. Thus optimizing (A33) is equivalent to

$$
\begin{aligned}
&\min_{\boldsymbol{\alpha}} D_{KL}(p^J(\theta) \parallel \frac{1}{Z'(\boldsymbol{\alpha})} p_{\boldsymbol{\alpha}}^2(\theta)) \\
\Leftrightarrow\ &\min_{\boldsymbol{\alpha}} \int p^J(\theta) \log \frac{p^m(\theta)}{p_{\boldsymbol{\alpha}}(\theta)} d\theta + \log \frac{\int p_{\boldsymbol{\alpha}}^2(\theta) d\theta}{\int p^m(\theta) p_{\boldsymbol{\alpha}}(\theta) d\theta} \\
\Leftrightarrow\ &\min_{\boldsymbol{\alpha}} \int p^J(\theta) \log \frac{p^m(\theta)}{p_{\boldsymbol{\alpha}}(\theta)} d\theta - \log \int p^m(\theta) p_{\boldsymbol{\alpha}}(\theta) d\theta.
\end{aligned}
\tag{A36}
$$

Denote the optimal solution for (A36) by $\boldsymbol{\alpha}^\dagger$. Then we can build the connection between $\boldsymbol{\alpha}^\dagger$ and $\hat{\boldsymbol{\alpha}}$ by

$$
\int p^J(\theta) \log \frac{p^m(\theta)}{p_{\boldsymbol{\alpha}^\dagger}(\theta)} d\theta - \log \int p^m(\theta) p_{\boldsymbol{\alpha}^\dagger}(\theta) d\theta \leq \int p^J(\theta) \log \frac{p^m(\theta)}{p_{\hat{\boldsymbol{\alpha}}}(\theta)} d\theta - \log \int p^m(\theta) p_{\hat{\boldsymbol{\alpha}}}(\theta) d\theta. \tag{A37}
$$

Since $\hat{\boldsymbol{\alpha}}$ minimizes $\int p^J(\theta) \log \frac{p^m(\theta)}{p_{\boldsymbol{\alpha}}(\theta)} d\theta$, which means $\int p^J(\theta) \log \frac{p^m(\theta)}{p_{\hat{\boldsymbol{\alpha}}}(\theta)} d\theta \leq \int p^J(\theta) \log \frac{p^m(\theta)}{p_{\boldsymbol{\alpha}^\dagger}(\theta)} d\theta$, we can get

$$
-\log \int p^m(\theta) p_{\boldsymbol{\alpha}^\dagger}(\theta) d\theta \leq -\log \int p^m(\theta) p_{\hat{\boldsymbol{\alpha}}}(\theta) d\theta, \tag{A38}
$$

or

$$
\int p^m(\theta) p_{\boldsymbol{\alpha}^\dagger}(\theta) d\theta \geq \int p^m(\theta) p_{\hat{\boldsymbol{\alpha}}}(\theta) d\theta, \tag{A39}
$$

which gives

$$
\int_{\theta \in S} p^m(\theta) p_{\boldsymbol{\alpha}^\dagger}(\theta) d\theta + \int_{\theta \in \bar{S}} p^m(\theta) p_{\boldsymbol{\alpha}^\dagger}(\theta) d\theta \geq \int_{\theta \in S} p^m(\theta) p_{\hat{\boldsymbol{\alpha}}}(\theta) d\theta + \int_{\theta \in \bar{S}} p^m(\theta) p_{\hat{\boldsymbol{\alpha}}}(\theta) d\theta. \tag{A40}
$$

Then we have

$$
\begin{aligned}
\int_{\theta \in S} p^m(\theta) p_{\boldsymbol{\alpha}^\dagger}(\theta) d\theta &\geq \int_{\theta \in S} p^m(\theta) p_{\hat{\boldsymbol{\alpha}}}(\theta) d\theta + \int_{\theta \in \bar{S}} p^m(\theta) p_{\hat{\boldsymbol{\alpha}}}(\theta) d\theta - \int_{\theta \in \bar{S}} p^m(\theta) p_{\boldsymbol{\alpha}^\dagger}(\theta) d\theta \\
&\geq \int_{\theta \in S} p^m(\theta) p_{\hat{\boldsymbol{\alpha}}}(\theta) d\theta - (\gamma_2 - \gamma_1).
\end{aligned}
\tag{A41}
$$

From Assumption 1 we have

$$
\frac{p^m(\theta^*) p_{\boldsymbol{\alpha}^\dagger}(\theta^*)}{p^*(\theta^*)} \geq \frac{p^m(\theta^*) p_{\hat{\boldsymbol{\alpha}}}(\theta^*)}{p^*(\theta^*)} - (\gamma_2 - \gamma_1), \tag{A42}
$$

which gives

$$KL_{\boldsymbol{\alpha}^\dagger} = -\log \frac{p_{\boldsymbol{\alpha}^\dagger}(\theta^*)}{p^*(\theta^*)} \leq -\log \left( \frac{p_{\hat{\boldsymbol{\alpha}}}(\theta^*)}{p^*(\theta^*)} - \frac{\gamma_2 - \gamma_1}{p^m(\theta^*)} \right) \leq -\log \frac{p_{\hat{\boldsymbol{\alpha}}}(\theta^*)}{p^*(\theta^*)} + \frac{\gamma_2 - \gamma_1}{p^m(\theta^*)}, \tag{A43}$$

or

$$KL_{\boldsymbol{\alpha}^\dagger} \leq KL_{\hat{\boldsymbol{\alpha}}} + \frac{\gamma_2}{C_2}. \tag{A44}$$

After combining with Theorem 1, we have

$$KL_{\boldsymbol{\alpha}^\dagger} \leq KL_{\boldsymbol{\alpha}^*} + \frac{2\gamma_2^2}{C_5} \log \frac{C_6}{\gamma_2} + \frac{\gamma_2}{C_2}. \tag{A45}$$

This means by optimizing our final objective (A33), the KL divergence $KL_{\boldsymbol{\alpha}^\dagger}$ is also bounded near the optimal value, which provides a theoretical justification of our algorithm.

### 1.4 Tips for Practitioners

In Section 2.4, we propose a two-stage algorithm, where we update the task weights with Langevin dynamics in the first stage, and then udpate the model with fixed task weights in the second stage. However, we find in practice that we can also find the similar task weights if we turn off the Langevin dynamics and directly sample from regular SGD. Therefore, we can further simplify the algorithm by removing the Langevin dynamics and merge the two stage, *i.e.*, update task weights and model parameters at the same time until convergence. This simplified version is summarized in Algorithm 1.

---

**Algorithm 1** ARML (simplified version)

---

**Input:** main task data $\mathcal{T}_m$, auxiliary task data $\mathcal{T}_{a_k}$, initial parameter $\theta_0$, initial task weights $\boldsymbol{\alpha}$
**Parameters:** learning rate of $t$-th iteration $\epsilon_t$, learning rate for task weights $\beta$

**for** iteration $t = 1$ to $T$ **do**
    $\theta_t \leftarrow \theta_{t-1} - \epsilon_t(-\nabla \log p(\mathcal{T}_m|\theta_{t-1}) - \sum_{k=1}^K \alpha_k \nabla \log p(\mathcal{T}_{a_k}|\theta_{t-1})) + \eta_t$
    $\boldsymbol{\alpha} \leftarrow \boldsymbol{\alpha} - \beta \nabla_{\boldsymbol{\alpha}} \|\nabla \log p(\mathcal{T}_m|\theta_t) - \sum_{k=1}^K \alpha_k \nabla \log p(\mathcal{T}_{a_k}|\theta_t)\|_2^2$
    Project $\boldsymbol{\alpha}$ back into $\mathcal{A}$
**end for**

---

## 2 Experimental Settings

For all results, we repeat experiments for three times and report the average performance. Error bars are reported with CI=95%. In our algorithm, the only hyperparameter is the learning rate $\beta$ of task weights. Specifically, we find the results insensitive to the choice of $\beta$. Therefore, we randomly choose $\beta \in [0.0005, 0.05]$, for a trade-off between steady training and fast convergence. We use PyTorch [19] for implementation.

### 2.1 Semi-supervised Learning

For semi-supervised learning, we use two datasets, CIFAR10 [11] and SVHN [17]. For CIFAR10, we follow the standard train/validation split, with 45000 images for training and 5000 for validation. Only 4000 out of 45000 training images are labeled. For SVHN, we use the standard train/validation split with 65932 images for training and 7325 for validation. Only 1000 out of 65392 images are labeled. Both datasets can be downloaded from the official PyTorch torchvision library (https://pytorch.org/docs/stable/torchvision/index.html). Following [18], we use WRN-28-2 as our backbone, *i.e.*, ResNet [7] with depth 28 and width 2, including batch normalization [9] and leaky ReLU [15]. We train our model for 200000 iterations, using Adam [10] optimizer with batch size of 256 and learning rate of 0.005 in first 160000 iterations and 0.001 for the rest iterations.

For implementation of self-supervised semi-supervised learning (S4L), we follow the settings in the original paper [23]. Note that we make two differences from [23]: (i) for steadier training, we use the model with time-averaged parameters [21] to extract feature of the original image, (ii) To avoid over-sampling of negative samples in triplet-loss [1], we only put a loss on the cosine similarity between original feature and augmented feature.

## 2.2 Multi-label Classification

For multi-label classification, we use CelebA [14] as our dataset. It contains 200K face images, each labeled with 40 binary attributes. We cast this into a multi-label classification problem, where we randomly choose one attribute as the main classification task, and other 39 as auxiliary tasks. We randomly choose 1% images as labeled images for main task. The dataset is available at `http://mmlab.ie.cuhk.edu.hk/projects/CelebA.html`. We use ResNet18 [7] as our backbone. We train the model for 90 epochs using SGD solver with batch size of 256 and scheduled learning rate of 0.1 initially and $0.1\times$ shrinked every 30 epochs.

## 2.3 Domain Generalization

Following the literature [2, 4], we use PACS [12] as our dataset for domain generalization. PACS consists of four domains (photo, art painting, cartoon and sketch), each containing 7 categories (dog, elephant, giraffe, guitar, horse, house and person). The dataset is created by intersecting classes in Caltech-256 [6], Sketchy [20], TU-Berlin [5] and Google Images. Dataset can be downloaded from `http://sketchx.eecs.qmul.ac.uk/`. Following protocol in [12], we split the images from training domains to 9 (train) : 1 (val) and test on the whole target domain. We use a simple data augmentation protocol by randomly cropping the images to 80-100% of original sizes and randomly apply horizontal flipping. We use ResNet18 [7] as our backbone. Models are trained with SGD solver, 100 epochs, batch size 128. Learning rate is set to 0.001 and shrinked down to 0.0001 after 80 epochs.