[Reviews · NeurIPS 2020]

Review 1

Summary and Contributions: This paper aims to address the supervised learning problems where the labelled data is scarce. The authors propose to use the auxiliary tasks to provide the additional supervision which is helpful to the main task of interest. The idea is based on the assumption there is a true prior which the optimal parameters for the main task is sampled from. The authors propose to form a surrogate prior based on the auxiliary tasks and expect that minimizing the KL divergence between the surrogate prior and the true prior can reduce the data requirement of the main task. Directly minimizing the KL divergence is intractable as the true prior is unknown. After some derivative, the authors propose a practical optimization objective that reduces the distance the gradients of the main task and weighted auxiliary task losses to learn the best weighting of auxiliary tasks. The proposed method has been verified on semi-supervised learning, multi-label classification and few-shot domain adaptation problems.

Strengths: 1. Casting the data-efficient learning to a multitask reweighting learning problem is interesting. 2. The proposed method sounds reasonable and can be applied broadly on semi-supervised learning, multi-label classification and few-shot domain adaptation problems. 3. The overall presentation of the paper is well organized.

Weaknesses: 1. L110 p^*(\theta) is 'covered' by p(T_m|\theta) is not clearly explained. 2. The proposed method is highly related to CosineSim[13] and OL_AUX[31]. However, the explanation/analysis of why the proposed method is better fit to minimum-data learning is not clear. 3. It would be nice if the authors have provided some analysis of the effect by adding varying the numbers (K) of auxiliary tasks for at least one same problem. 4. Line 239-242 the text here is confusing. According to Algorithm 1, the main task loss will be minimized after learning the auxiliary task weights \alpha. Why does the text say the extra labels are only used for reweighing? So the authors cut off the main task learning here? However, even though the main task learning may be cut off after learning the auxiliary task weights. The task reweighting is learned with the main task classification signal. Thus, the statement "the improvement completely comes from task reweighting" is over claimed. The comparisons between JT/FADA and JT+ARML/FADA+ARML are fairer to verify the effectiveness of ARML, which shows ARML can only give marginal improvement over the vanilla JT/FADA methods in Table 3. 5. It would be good if the authors have also focused on some non-classification problems in their paper.

Correctness: The description of the proposed method is relatively clear. However, why the proposed method is more advanced than the previously related contributions is not clearly discussed.

Clarity: Most part is well written.

Relation to Prior Work: The motivation/angle of the proposed method is different from previous contributions. However, the empirical implementation of the proposed method is highly related to the existing works, eg, CosineSim[13] and OL_AUX[31]. The explanation of why the contribution is better than the previous ones is not clear.

Reproducibility: No

Additional Feedback: L110 p^*(\theta) is 'covered' by p(T_m|\theta). This needs more clear explaination. Thanks for the feedback of the authors. Now, after reading the rebuttal of the authors and the reviews from the other reviewers, I think most of my concerns have been addressed and agree this paper has its merit. Now I would like to raise my rating.


Review 2

Summary and Contributions: In this paper, they introduce a method for adaptively re-weight the auxiliary task importance in order to select more important tasks (high quality tasks which are more related to the target task). The optimization problem is minimizing the distance between the gradient of main and auxiliary loses. Their experiments on several settings such as semi-supervised, few-shot, and domain adaptation are interesting and promising.

Strengths: - This paper is very well-written and easy to follow. Especially they have a simplified version fo they theorem. (I appreciate that) - The way they approached to the learning with auxiliary tasks problem is innovative. That they formulated a surrogate prior for the main task using auxiliary tasks’ likelihood. This is a big contribution to me. - They performed extensive experiments considering different research areas that could be connected to theirs such as few-shot, domain adaptation, and self-supervised learning methods.

Weaknesses: - I think it is informative to have some explanation in the Algorithm section. Something that is not clear to me is how they sample from auxiliary data during training. Is that correct that during first epochs, the estimation of relatendness is not much accurate since we are in the primary stages of training, but after some epochs, more related auxiliary tasks are selected. - There is no discussion about how efficient is the proposed method (since objective function is the distance between the gradient of two losses).

Correctness: yes

Clarity: yes

Relation to Prior Work: yes

Reproducibility: Yes

Additional Feedback: I read the rebuttal, I keep my score.


Review 3

Summary and Contributions: A method to adaptively reweight auxiliary tasks in order to reduce the data required on the main task is proposed. The key assumption is that high-quality prior can reduce the data requirement, and the parameter distribution induced by the auxiliary tasks' likelihood is formulated as surrogate prior to the main task. Experimental results show that the proposed method leads to a significant gain over unsupervised and few-shot baselines using very little labeled data.

Strengths: As data-efficient learning has always been important to the community, the addressed problem of utilizing auxiliary task matters to the NeurIPS. The theoretical groundings and claims look very sound to me, and formulations are straightforward to follow and verify. The proposed algorithm ARML is general and widely applicable. The algorithm looks relatively easy to reproduce, and the supplemental material and the provided code helped to understand. One of the most interesting experiments is in Table 2. The proposed method seems to work the best under the setting of multi-label classification. If more experiments can be provided in this direction, it would be helpful.

Weaknesses: While the proposed algorithm is very interesting, I still have a question of how general this approach can be applied. In practice, we still need to find ways to decide which tasks can be associated. Have you found the cases when auxiliary tasks can theoretically or empirically harm the main task? Do we have a way to prevent that harm? Looking into Table 1., VAT + EntMin, an existing work, seems to work much better than the proposed method. Can you explain this? Similar to the original S4L paper, have you tried the MOAM approach(Mix of All Models)?

Correctness: The claims and methods are correct and the experimental results look correct.

Clarity: The paer is well written.

Relation to Prior Work: Relevant work is appropriately discussed.

Reproducibility: Yes

Additional Feedback:


Review 4

Summary and Contributions: This paper considers the problem of learning a main task along with several auxiliary tasks. Self-supervised learning (SSL) is one such example. The authors propose a method to learn the weights of the losses of the auxiliary tasks such that the model can perform well on the main task even when it has very limited amount of labels. The rationale behind the proposed method is quite novel and interesting. The algorithm that the authors arrive at is simple and clear. The authors conducted several experiments in both the SSL setting with a small amount of main task labels and the few-shot learning setting where only a few labels are available for the main task. The experiment results showed the effectiveness of the proposed method.

Strengths: Formulating learning auxiliary task weights as prior matching is quite novel. The techniques employed such as Langevin dynamics and score matching are clever. The resulting algorithm is relatively simple.

Weaknesses: The theoretical justification for the approximation steps is not entirely convincing, esp. the justification behind switching sampling probability to solve P1. [UPDATED] Given the author feedback, I think the theoretical justification based on Theorem 1 and that in the Appendix to support eq. 11 is valid.

Correctness: The logic behind using Theorem 1 to justify the switch of sampling probability is flawed. Eq. (8) showed that the solution to the approximate problem $\hat{\alpha}$ forms a lower bound to KL-divergence involving the ideal solution $\alpha^*$, which is the quantity that we really want to minimize. It doesn't make a lot of sense to minimize a lower bound of the objective that you would like to minimize. For example, in both EM and variational inference, we're trying to maximize a lower bound of the objective that we really would like to maximize. So in this case, the justification will be more plausible if the solution to the approximate problem can be shown to form an upper bound. [UPDATED] Given the author feedback, my previous question raised above was indeed due to misunderstanding. To make the purpose of Theorem 1 clearer, I recommend the authors to add a lower bound (by the definition of $\alpha*$), i.e., D_{KL}(p* \| p_{\alpha*} \leq D_{KL}(p* \| p_{\hat{\alpha}} \leq ... so that it becomes clear that $\hat{\alpha}$ can deliver an objective closer to the optimal one.

Clarity: Overall this paper is clear. However, the concept of "true prior." The authors didn't offer a clear definition for it. It's a little dissatisfying to see an undefined vague concept in formal theoretical derivation such as Theorem 1. If an oracle can offer the true parameter, then the true prior should just be a Dirac delta on that true parameter. Plus, "true" implies "unique." It seems to me that the target prior that the authors try to get close to can simply be any prior around the best parameter (assuming infinite amount of data) with small support.

Relation to Prior Work: The related work section is clearly written.

Reproducibility: Yes

Additional Feedback: Details: 1) line 5. "a crucial" 2) line 53. you might want to make it clear that "very little labeled data" is for the main task since auxiliary tasks also need labeled data (just easier to get if they're self-supervised). 3) line 123. "out of S" -> "outside S" 4) Section 3.1. I see no mentioning of the model architecture used. 5) Table 1. I see no mentioning of the number of runs to get the standard deviation. 6) Section 3.1. It would be interesting to show the learned weights (or weight distribution from multiple runs) per task for illustration purposes. [UPDATED] In the author feedback, the authors mentioned that they did 3 runs to obtain the standard deviation of the test error presented in Table 1. 3 runs don't give me much confidence in the obtained standard deviation. I think more runs, e.g., 20, are needed to make the results more reliable.

[Author Response · NeurIPS 2020]

Table 1: Starting from one auxiliary task (Exemplar-MT), we keep increasing the number of auxiliary tasks from one to four by adding one auxiliary task at a time in the order of {Rotation, VAT, EntMin}.

| K | 1 (+Exemplar-MT) | 2 (+Rotation) | 3 (+VAT) | 4(+EM) |
|---|---|---|---|---|
| CIFAR-10 | 17.02 | 13.68 | 12.47 | **11.80** |
| SVHN | 9.65 | 5.89 | 5.00 | **4.70** |

Table 2: Effect of ARML on harmful tasks.

| Exemplar-MT | + Rotation (harmful) |
|---|---|
| 17.02 | **16.95** (w/ ARML) |
| | 83.76 (w/o ARML) |

We would like to thank all the reviewers for writing the insightful comments, especially during this difficult time.

R1.1 - **L110 "$p^*(\theta)$ is 'covered' by $p(\mathcal{T}_m|\theta)$"** is not clearly explained: Thanks! By 'covered', we mean $p(\mathcal{T}_m|\theta)$ has
high density both in the support set $S$ of $p^*(\theta)$, and in some regions outside $S$. We will clarify this in the final version.

R1.2 - **Our contribution and difference from previous works**: Thanks. We proposed ARML for auxiliary task
reweighting so that the least data is required to find the true parameter. Although ARML and some previous algorithms
(*e.g.* CosineSim, OL_AUX) both update task weights based on the similarity between main/auxiliary task gradient, the
previous algorithms mainly focus on the **training loss**. For example, CosineSim discards the harmful tasks so that the
training loss of the main task will not increase in one step. OL_AUX updates the task weights so that the training loss
decreases the fastest. However, lowering training loss may cause overfitting, especially when training data is scarce.
In contrast, ARML is guaranteed to find a good prior so that the least data is required to find the parameter which
generalizes the best (*i.e.* having the lowest **test error**). The superiority of ARML is verified in the experiments.

R1.3 - **Ablation on number of auxiliary tasks**: Thanks. We ablate on the number of auxiliary tasks in SSL, and
observe the results in Table 1. The error rates decreases when each new task is added.

R1.4 - **Line 239-242 the text here is confusing**: Thanks. For domain generalization baseline, the model is trained only
with source domain (auxiliary) data. In 'Baseline + ARML', for fair comparison, we stick to the same training process,
*i.e.*, updating $\theta$ using only auxiliary loss (NOT using the main task loss), and the main task data is used to adjust the
task weights $\alpha$ which brings the result improvement. We will add more elaboration on this in the final version.

R1.5 - **Non-classification experiments**: Thanks for the suggestion! We will try other tasks, *e.g.*, reinforcement learning.

R2.1 - **More explanation in the Algorithm section**: Thanks! (1) We use mini-batch sampling for main/auxiliary task
data. In each mini-batch, we sample the same amount of two kinds of data. (2) Yes, during early epochs the sampling
process is still warming up, and after that the estimation is more accurate. We will add more details in the section.

R2.2 - **Discussion on the efficiency of the proposed method**: Thanks. The only overhead of ARML is the update of
$\alpha$. This has little extra cost because the main/auxiliary task gradients are already calculated when updating $\theta$, and we
only have to calculate the gradient w.r.t. $\alpha$. We observed a $< 10\%$ extra cost for PyTorch, and nearly no extra cost for
TensorFlow (because of the different autograd strategies of the two frameworks). The efficiency is similar to previous
algorithms (*e.g.* OL_AUX), and much higher than GridSearch. We will update this in the final version.

R3.1 - **More experiments on multi-label classification**: Thanks! We will add more experiments including 1) analyzing
if the learned face attribute relationship aligns with human's intuition, 2) varying the number of auxiliary attributes.

R3.2 - **Can ARML prevent harmful tasks?**: Thanks! Ideally, ARML can discard a harmful task by lowering its
weight to 0. For verification, we turn the Rotation loss in S4L into its negative, making it a harmful task (which means
when training we are actually increasing the loss). Meanwhile we keep the Exemplar-MT loss unchanged. We find that
ARML lowers the weight of Rotation to 0, getting the same error rate as when only Exemplar-MT is applied (Table 2).

R3.3 - **Why does VAT + EntMin works better than S4L + ARML? Have you tried MOAM?**: Since ARML already
finds the optimal task weights, the performance is mainly limited by the auxiliary tasks themselves. For example, if we
further add VAT & EntMin as auxiliary tasks (which makes it MOAM), then ARML surpasses SOTA (R1.3 & Table 1).

R4.1 - **Justification behind switching sampling probability (Theorem 1)**: Thanks. We think there may be a small
misunderstanding, because Theorem 1 does not mean that we are minimizing a lower bound of the true objective. Our
true objective is to find the optimal task weights $\alpha^\star$ so that $D_{KL}(p^* \parallel p_{\alpha^\star})$ is the smallest. Theorem 1 states that,
if we choose the weights $\hat{\alpha}$ (from the optimization of the surrogate objective), then the corresponding true objective
$D_{KL}(p^* \parallel p_{\hat{\alpha}})$ is also very small, *i.e.*, upper-bounded near the optimal value $D_{KL}(p^* \parallel p_{\alpha^*})$.

R4.2 - **The concept of "true prior"**: This is a good point! The "true prior" is an objective prior reflecting the
randomness of the environment. For example, for image classification problems with pictures taken in different places,
the model parameter should stay similar to extract the same features, while bearing some slight changes (*e.g.* different
image statistics in BN). The "true prior" can also be regarded as a Dirac delta if not considering such randomness.

R4.3 - **Details**: Thank you! We will correct the typos in the final version. 4) WRN-28-2 is used for SSL experiments,
and ResNet18 is used for others. 5) We take three runs for each result. 6) Thanks. We will add the illustrations.

[Meta-Review · NeurIPS 2020]

This paper is well-written. All the reviewers agree that the proposed method is reasonable and interesting. The authors' feedback addresses most of the concerns raised by reviewers.